# MinVIS: A Minimal Video Instance Segmentation Framework without Video-based Training

**De-An Huang**
NVIDIA
deahuang@nvidia.com

**Zhiding Yu**
NVIDIA
zhidingy@nvidia.com

**Anima Anandkumar**
Caltech, NVIDIA
anima@caltech.edu

## Abstract

We propose MinVIS, a minimal video instance segmentation (VIS) framework that achieves state-of-the-art VIS performance with neither video-based architectures nor training procedures. By only training a query-based image instance segmentation model, MinVIS outperforms the previous best result on the challenging Occluded VIS dataset by over 10% AP. Since MinVIS treats frames in training videos as independent images, we can drastically sub-sample the annotated frames in training videos without any modifications. With only 1% of labeled frames, MinVIS outperforms or is comparable to fully-supervised state-of-the-art approaches on YouTube-VIS 2019/2021. Our key observation is that queries trained to be discriminative between intra-frame object instances are temporally consistent and can be used to track instances without any manually designed heuristics. MinVIS thus has the following inference pipeline: we first apply the trained query-based image instance segmentation to video frames independently. The segmented instances are then tracked by bipartite matching of the corresponding queries. This inference is done in an online fashion and does not need to process the whole video at once. MinVIS thus has the practical advantages of reducing both the labeling costs and the memory requirements, while not sacrificing the VIS performance. Code is available at: `https://github.com/NVlabs/MinVIS`

## 1 Introduction

Video instance segmentation (VIS) aims to simultaneously detect, segment, and track object instances in videos [1]. The requirement to accurately track object instances through an entire video makes VIS much more challenging than image instance segmentation. Most of the early approaches for VIS build on image instance segmentation models, and process videos on a *per-frame* basis [1, 2]. The segmented object instances for each frame are then matched temporally with a post-processing step. This post-processing step often involves manually designed heuristics that do not generalize well to challenging scenarios like occlusions and large appearance deformations.

Recent VIS works address this issue by taking a *per-clip* approach, where the spatial-temporal volume of a video is processed as a whole to directly predict the spatial-temporal mask for each object instance [3–5]. Many of these end-to-end VIS approaches are built upon the recent advances of Transformers for end-to-end object detection [6]. Given learned embeddings called *queries*, Transformers process the queries jointly with the input video using cross-attention, so that each of the processed queries can be used to predict the spatial-temporal mask for an object instance in the video.

While these per-clip methods have led to considerable improvements for VIS, using attention to process the whole video, especially longer ones, requires large memory and computation. It is also not straightforward to adapt per-clip methods from offline to online processing to reduce the computational requirements. This limits their practical application, and maintaining the effectiveness of these per-clip methods while improving their efficiency remains an active research direction [7, 8].

36th Conference on Neural Information Processing Systems (NeurIPS 2022).

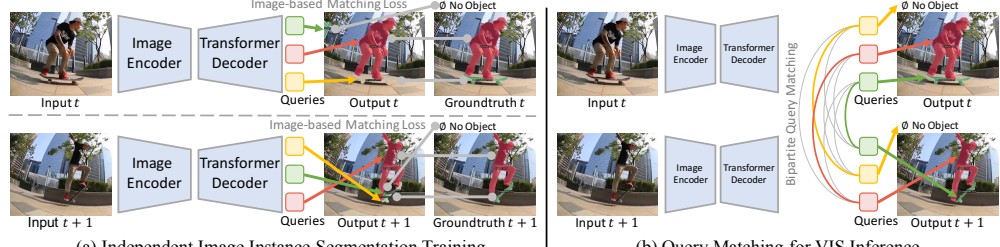

Figure 1: (a) MinVIS trains a query-based image instance segmentation model (Image Encoder + Transformer Decoder) using each frame independently. (b) During inference, the trained image instance segmentation model is used for video instance segmentation by bipartite matching of query embeddings across frames. MinVIS does not require further manually designed heuristics for tracking.

Another limitation for existing VIS methods is the requirement on annotation. Annotating object instance masks for each video frame is prohibitively expensive at scale. While there have been works that alleviate this annotation requirement through weak supervision or image-based annotation, there is still a significant performance gap compared to state-of-the-art fully-supervised methods [9, 10].

**Our Approach.** We simultaneously address both of the aforementioned challenges of computational and labeling costs by showing that we can achieve state-of-the-art VIS performance by only training a query-based *image* instance segmentation model. During inference, MinVIS first applies the query-based image instance segmentation to video frames independently. The segmented instances are then associated by bipartite matching of the corresponding queries. MinVIS processes each frame independently in an online fashion and does not need to process the whole video at once. MinVIS does not use any video-based training procedure and thus does not need annotations for all the frames in a video. Our contributions are summarized below:

1. We show that video-based architecture and training are not required for competitive VIS performances. MinVIS outperforms previous state-of-the-art on YouTube-VIS 2019 and 2021 datasets by 1% and 3% AP while only training an *image* instance segmentation model.

2. We show that image instance segmentation models capable of segmenting occluded instances are also well suited to track occluded instances in videos in our framework. MinVIS outperforms its per-clip counterpart by over 13% AP on the challenging Occluded VIS (OVIS) dataset, which is over 10% improvement compared to the previous best performance on the dataset.

3. Our image-based approach allows us to significantly sub-sample the required segmentation annotations in training without any change to the model. With only 1% of labeled frames, MinVIS outperforms or is comparable to fully-supervised state-of-the-art approaches on all three datasets.

Our key observation is that queries trained to be discriminative between intra-frame object instances are temporally consistent and can be used to track instances without being trained with video-based loss functions. MinVIS achieves this by requiring its image instance segmentation model to generate masks by convolving query embeddings with features of the whole input image, including regions of other object instances. A query is thus trained to only have high responses on features of its corresponding instance. Other query embeddings should instead have low responses on these features because instance masks are non-overlapping. This design encourages the query embeddings for different instances in a frame to be well-separated. On the other hand, the query embeddings that segment the same instance from two consecutive frames still need to be similar enough since the instance's image features to be convoluted do not change drastically between frames. This leads to temporally consistent query embeddings for tracking without the need of video-based training.

MinVIS thus has the following design for inference: We first apply a query-based image instance segmentation model on video frames independently. The segmented instances are then associated between frames by bipartite matching of the corresponding query embeddings. This post-processing step does not need any additional heuristics based on mask overlaps or classification scores as in previous works [1, 11]. This is because query embeddings already contain these information to track the instances. An overview of MinVIS's training and inference is shown in Figure 1.

Since video frames are treated as independent images to train MinVIS, there is no requirement to annotate all the frames in a video for training. This allows us to significantly sub-sample and reduce the annotation without any change to our model. We find that on YouTube-VIS 2019/2021 datasets [1], where there are less variations between video frames, using only 1% of labeled frames leads to less than 3% drop in AP for MinVIS.

We further evaluate MinVIS on the Occluded VIS (OVIS) dataset [12]. One common critique of per-frame approaches is that their tracking heuristics based on mask overlaps would not work when there are heavy occlusions. This is not the case for MinVIS, as we do not use any manually designed heuristics. We show that our query-matching approach generalizes to occluded scenarios. MinVIS with Swin Transformers backbone [13] achieves 39.4% AP on OVIS, which is over 10% improvement from the previous best result on the dataset [14]. We further show that our image-based strategy leads to easier and better learning on OVIS. MinVIS outperforms its per-clip counterpart by over 13% AP.

## 2    Related Work

**Video Instance Segmentation.** Per-frame approaches for VIS process each frame independently and later track instances by post-processing. MaskTrack R-CNN [1] adds a tracking head to Mask R-CNN [15] for VIS. MaskProp [16] instead adds a mask propagation head to propagate object instance masks. CrossVIS [2] uses crossover learning to improve instance representation across video frames. QueryTrack [11] adds a contrastive tracking head to QueryInst [17] for instance association. Concurrent work IDOL [18] shows that per-frame models can still outperform per-clip models. Our approach also builds on image instance segmentation models, but unlike previous approaches, we need neither additional parameters nor additional losses to apply to VIS. Our query embeddings from image instance segmentation can directly be used for tracking without video-based training.

Recent per-clip approaches build on the success of Detection Transformer (DETR) [6]. VisTR [4] adopts the query-based approach of DETR to VIS, and there has been several follow-up works, such as Mask2Former-VIS [3] and SeqFormer [5]. One limitation of these approaches is the need to process the whole video at once. IFC [7] reduces the overhead of temporal message passing by using memory tokens. TeViT [8] uses a parameter-shared self attention to efficiently model temporal contexts. We also use a query-based approach, but instead of using cross-attention to process the whole video, we process each frame independently while not losing VIS performance. Our use of queries to associate instances is also related to other works that build on DETR for tracking in related fields. For example, MOTR [19] and TrackFormer [20] use identity preserving track queries for multi-object tracking (MOT).

**Reducing Supervision for Video Instance Segmentation.** Annotating instance masks for each video frame can be prohibitively expensive. Compared to video object segmentation [21–23] and image instance segmentation [24–27], there have been less works on reducing supervision for VIS [28]. FlowIRN [10] extends IRN [24] with motion and temporal consistency cues to have a weakly-supervised VIS framework that only requires classification labels. SOLO-Track [9] learns to track instances without video annotations. It uses instance contrastive learning on SOLO [29] to learn grid cell embeddings for instance tracking. We make the same observation that disciminating between instances within frames is beneficial or even sufficient for instance tracking. However, unlike our query-based association, the grid cell embeddings still need threshold-based post-processing and additional loss functions to better handle birth and death of objects.

## 3    Method

MinVIS is a minimal VIS framework that does not require video-based training and thus can be easily applied to real-world applications that only have sparse image instance segmentation annotations. MinVIS is a two stage approach: (1) image instance segmentation on each frame independently, (2) associating instances between frames by matching queries. We will first discuss the image instance segmentation architecture in MinVIS. We will then discuss the temporal association of object instances. Finally, we will discuss training and reducing supervision for MinVIS.

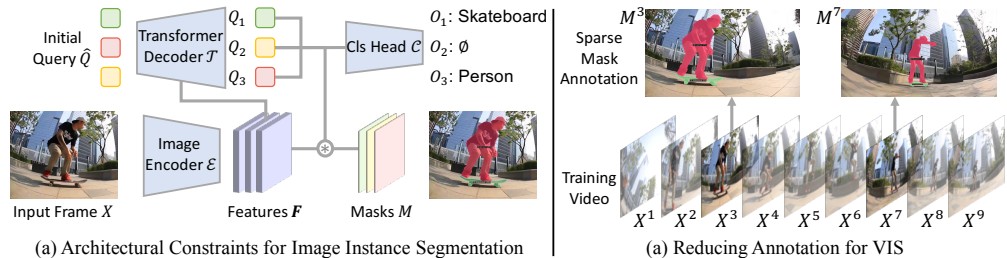

| (a) Architectural Constraints for Image Instance Segmentation | (a) Reducing Annotation for VIS |

Figure 2: (a) MinVIS's main architectural constraint is to require the segmentation masks $M$ be generated by convolving the query embeddings $Q$ with the final feature map $F_{-1}$. This makes the query embeddings discriminative between each instances. (b) MinVIS's image-based approach allows direct annotation subsampling of training videos without any modification to the model.

## 3.1 Image Instance Segmentation Architecture for VIS

MinVIS builds on the query-based transformer architectures for detection and segmentation [6, 17, 30, 31], which has the following main components: (1) *Image Encoder* that learn to extract features from input images. (2) *Transformer Decoder* that processes the outputs of Image Encoder to iteratively update the query embeddings. (3) *Prediction Heads* that use the final query embeddings to predict desired outputs (*e.g.,* segmentation masks, class labels). The queries play an important role for the success of such end-to-end pipeline for set prediction with unknown number of outputs. The number of queries are selected as the maximum number of output instanes of the model. During inference, a subset of queries predict $\varnothing$ outputs to dynamically adjust the number of valid outputs.

An overview of MinVIS's image instance segmentation is shown in Figure 2(a). Given an image $X \in \mathbb{R}^{H,W}$, the Image Encoder $\mathcal{E}$ extracts a set of features $\boldsymbol{F} = \mathcal{E}(X)$ from the image. $\boldsymbol{F} = \{F_0 \dots F_{-1}\}$ is a sequence of multi-scale feature maps $F_i \in \mathbb{R}^{H_i,W_i,C_i}$. $F_{-1}$ denotes the final output of $\mathcal{E}$. The $N$ initial query embeddings $\hat{Q} \in \mathbb{R}^{N,C}$ are learnable parameters, where $N$ is a large enough number of outputs. The Transformer Decoder $\mathcal{T}$ then take both $\boldsymbol{F}$ and $\hat{Q}$ to iteratively obtain $Q = \mathcal{T}(\boldsymbol{F}, \hat{Q}), Q \in \mathbb{R}^{N,C}$. While most recent works focus on the design of $\mathcal{T}$ to better process $\boldsymbol{F}$ for $Q$, MinVIS's architectural constraints are on the Prediction Heads. There are two outputs for each instance: classification and segmentation mask. The classification scores $O = \mathcal{C}(Q), O \in \mathbb{R}^{N,K}$ for $K$ classes are the output of Classification Head $\mathcal{C}$, and $Q$ should contain the class information for each instance. For segmentation masks $M \in \mathbb{R}^{N,H,W}$, MinVIS requires that $M$ be generated by convolving the query embeddings $Q$ with the final feature map $F_{-1}$. The shape for $F_{-1}$ is thus $H, W, C$. We have $M = \sigma(Q * F_{-1})$, where $\sigma(\cdot)$ is the sigmoid function.

The constraint to have $Q$ convolve with the whole feature map $F_{-1}$ is important for MinVIS. Consider two queries $Q_i$ and $Q_j$ that corresponds to two distinct object instances and thus non-overlapping masks. This formulation ensures that $Q_i$ should only have high inner products with features in $F_{-1}$ that are covered by the mask of instance $i$. Since the instance masks are non-overlapping, $Q_j$ should instead have low inner products with these features. This implicitly constrains the query embeddings to be discriminative between each other. On the other hand, if we apply this pipeline to two consecutive frames $X^t$ and $X^{t+1}$. Then $Q_i^t$ should still have higher inner product with $Q_i^{t+1}$ compared to $Q_j^{t+1}$. This is because $Q_i^{t+1}$ and $Q_j^{t+1}$ are also discriminative between each other, while $Q_i^t$ and $Q_i^{t+1}$ both need to have high inner products with features of the same instance, which do not change drastically between consecutive frames. We visualize our learned query embeddings in Figure 3. Each plot is for a video. Query embeddings belonging to the same instance (from different frames) have the same color. These embeddings are already grouped by instances without any video-based training. Further details are in Section 4.2.

While not all image instance segmentation models satisfy our architectural constraints (*e.g.,* ROI-based architectures), we believe these are rather flexible designs that are compatible with various query-based instance segmentation models. We use Mask2Former [30] in this work. The Image Encoder $\mathcal{E}$ includes both the backbone and the pixel decoder of Mask2Former. We also find that having fully-connected layers to further process $Q$ before convolution is beneficial to the performance.

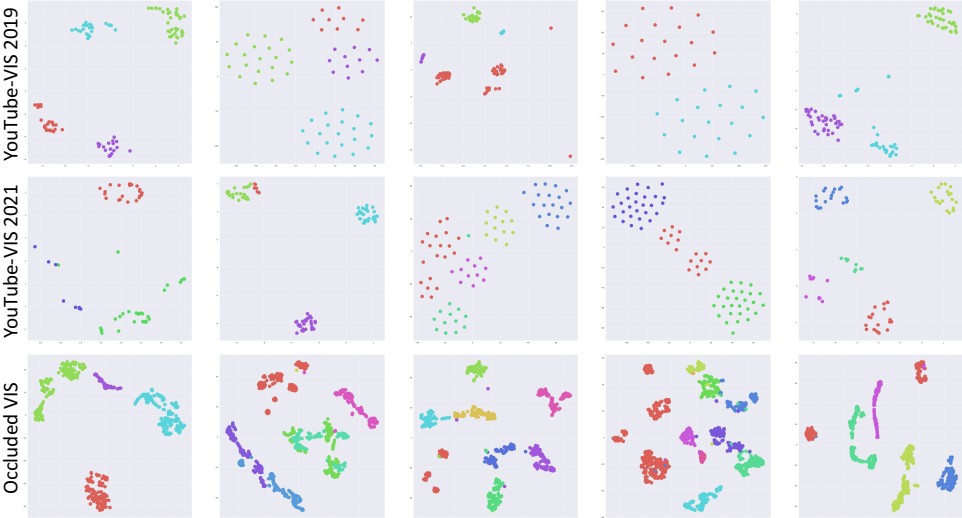

Figure 3: Visualizing our learned query embeddings with only image-based training. Each plot is for a video, and query embeddings of the same instance (from different frames) have the same color. Query embeddings are already grouped into clusters by instance without any video-based training.

## 3.2 Tracking by Query Matching

MinVIS is a per-frame two-stage approach, which requires a post-processing step to temporally associate instances. This post-processing often involves heuristics like mask overlaps, which does not generalize well to scenarios with heavy occlusions. Unlike previous two-stage approaches, MinVIS associate instances solely based on the query embeddings $Q$. Given two consecutive frames $X^t$ and $X^{t+1}$. We have $Q^t = \mathcal{T}(\boldsymbol{F}^t, \hat{Q})$, where $\boldsymbol{F}^t = \mathcal{E}(X^t)$, and similarly for $Q^{t+1}$. $Q_i^t$ is the query embedding for instance $i$. Tracking in MinVIS is done by the assignment of applying the Hungarian algorithm on a score matrix $S$, where $S_{ij} = cos(Q_i^t, Q_j^{t+1})$. $cos(\cdot, \cdot)$ is the cosine similarity.

This approach is less affected by occlusions because each instance is represented by a query that does not have a spatial extent. In addition, we do not need heuristics to handle the birth and death of object instances in this framework. Since the number of queries is larger than the number of instances, there are queries that produce empty masks. The death of an object instance happens when its embedding is matched to such a null query. On the other hand, the birth of an instance is correctly handled if the matched query embeddings have been null before the actual birth of the object instance.

## 3.3 Training with Less Supervision for VIS

Since the matching process does not need training, only the image instance segmentation model needs to be trained. There are two outputs of the model: classification scores $O \in \mathbb{R}^{N,K}$ and segmentation masks $M \in \mathbb{R}^{N,H,W}$ for $N$ queries, $K$ object classes, and image size $H, W$. We can process the groundtruth video instances to groundtruth image instances $O^* \in \mathbb{R}^{L,K}$ and $M^* \in \mathbb{R}^{L,H,W}$, where $L$ is the number of groundtruth instances ($N >> L$) and $O_j^*$ is a one-hot vector of groundtruth class. Given a loss function $\mathcal{L}(O_i, M_i, O_j^*, M_j^*)$ between predicted instance $i$ and groundtruth instance $j$, we first use bipartite matching to find the assignments between predicted and groundtruth instances that minimize the overall loss function, and train on those matched predictions with the loss function.

More specifically, there are two terms in the loss function: $\mathcal{L}_{cls}$ and $\mathcal{L}_{mask}$. We use cross entropy loss for $\mathcal{L}_{cls}$ and binary cross entropy plus dice loss [32] for $\mathcal{L}_{mask}$ as in previous works [30]. Both terms are purely image-based. The groudtruth video instances are first processed to instances for each frame independently. Therefore, even if there are only sparse frames labeled with instance, we can still train our model with the annotated frames. This provide a straightforward way to reduce the supervision for VIS. Figure 2(b) shows the annotation sub-sampling to reduce supervision.

# 4 Experiments

**Datasets.** We evaluate MinVIS on three datasets: YouTube-VIS 2019/2021 [1] and Occluded VIS (OVIS) [12]. The YouTube-VIS datasets contain 40 object classes. YouTube-VIS 2019 contains 2238/302/343 videos for training/validation/testing, while YouTube-VIS 2021 expands the dataset to 2985/421/453 videos for training/validation/testing, and includes higher quality annotations. OVIS has 25 object classes and contains 607/140/154 for training/validation/testing. While the number of videos is smaller, OVIS has more objects per frame, and the videos are also longer. This leads to more annotated instance masks compared to the YouTube-VIS datasets. In addition, OVIS also has much higher Bounding-box Occlusion Rate (0.22 v.s. 0.06/0.07) compared to the YouTube-VIS datasets, which indicates heavier occlusions between object instances.

**Metrics.** We follow previous works and use Average Precision (AP) and Average Recall (AR) as evaluation metrics [1]. AP is computed based on 10 intersection-over-union (IoU) thresholds from 50% to 95% with 5% increment. The reported AP and AR are first computed for each object class and then averaged over all classes. All three datasets have public evaluation servers.

**Baselines.** We focus on results using ResNet50 and Swin-L backbones. ResNet50 is still the most widely used backbone for VIS, while Swin-L gives the best performances. Not all methods report both backbones on all three datasets. We include results that are available. For YouTube-VIS datasets, we include recent state-of-the-art results from SeqFormer [5], TeViT [8], and Mask2Former-VIS [3]. These are all Transformer-based per-clip approaches as this paradigm has been recently dominating the field. On the other, out of of these methods, only TeViT is applied to OVIS. Therefore, we further compare to CMaskTrack R-CNN [12], CrossVIS [2], and STC [33]. These are all methods that allow online processing. Even TeViT uses a near online inference for OVIS [34]. This is because OVIS has longer videos that would lead to out-of-memory for most of the per-clip approaches.

Out of all the baselines, Mask2Former-VIS [3] is the most related to MinVIS, as MinVIS is built on Mask2Former in this work. Mask2Former-VIS thus can be seen as the per-clip version of ours and is an important baseline for comparison. Therefore, we further apply Mask2Former-VIS on the OVIS dataset. Due to memory constraints, the videos in OVIS are first split into clips of length 30. We use the same post-processing as MinVIS to merge the outputs from these clips.

**Implementation Details.** Unless otherwise noted, our hyper-parameters follow Mask2Former-VIS [3]. All models are pre-trained with COCO instance segmentation [35]. For OVIS, we use the same hyper-parameters as YouTube-VIS 2019 except training for $10k$ iterations instead of $6k$. For training losses, the weights are 5.0 for $\mathcal{L}_{mask}$ and 2.0 for $\mathcal{L}_{cls}$. All results of MinVIS are averaged over 3 random seeds. We sub-sample training to X% by uniformly sampling frames in the video. We set a minimum of 1 frame per video. Since YouTube-VIS datasets often have videos less than a hundred frames. Our 1% results are better seen as one frame per video results for YouTube-VIS.

## 4.1 Main Results

**YouTube-VIS 2019.** The results for YouTube-VIS 2019 are shown in Table 1. MinVIS achieves highest AP and most other metrics for both ResNet-50 and Swin-L backbones. SeqFormer shows that it is beneficial to jointly train with images from COCO [35] that contain YouTubeVIS categories (+C80k in table). TeViT proposes messenger shift transformer (MsgShifT) that are as efficient as ResNet backbones, while improving the VIS performances. Our ResNet-50 results match or outperform their results without further modifications. Compared to the state-of-the-art Mask2Former-VIS, which can be seen as the per-clip approach to apply Mask2Former to VIS, MinVIS consistently outperforms by around 1% for both backbones. MinVIS with X% means sub-sampling the annotated frames for each video in training. Since there are less temporal variations for videos in YouTube-VIS 2019, MinVIS with 1% of training frames only reduces AP by 2.6%. This significantly reduces the annotation effort while not sacrificing much performance.

**YouTube-VIS 2021.** The results for YouTube-VIS 2021 are shown in Table 2. On this more challenging dataset, the performance improvements for MinVIS increase compared to YouTube-VIS 2019. Without better backbone like TeViT and additional training data like SeqFormer, our ResNet-50 results outperform by a large margin for all metrics. This is the also case for Swin-L. MinVIS outperforms previous state-of-the-art Mask2Former-VIS by 2.7%. By using only 1% of training frames, MinVIS's AP decrease by only 2.4%, which means that our 1% result still outperforms

Table 1: YouTube-VIS 2019 results. C80k indicates joint training with COCO images that have YouTube-VIS categories. MinVIS with X% means sub-sampling the annotated frames in training.

| Method | Backbone | Training | AP | $AP_{50}$ | $AP_{75}$ | $AR_1$ | $AR_{10}$ |
|---|---|---|---|---|---|---|---|
| TeViT [8] | R50 | Full | 42.1 | 67.8 | 44.8 | 41.3 | 49.4 |
| TeViT [8] | MsgShifT | Full | 46.6 | **71.3** | 51.6 | 44.9 | 54.3 |
| SeqFormer [5] | R50 | Full | 45.1 | 66.9 | 50.5 | 45.6 | 54.6 |
| SeqFormer [5] | R50 | Full+C80k | **47.4** | 69.8 | 51.8 | 45.5 | 54.8 |
| Mask2Former-VIS [3] | R50 | Full | 46.4 | 68.0 | 50.0 | – | – |
| MinVIS | R50 | Full | **47.4** | 69.0 | **52.1** | **45.7** | **55.7** |
| TeViT [8] | Swin-L | Full | 56.8 | 80.6 | 63.1 | 52.0 | 63.3 |
| SeqFormer [5] | Swin-L | Full+C80k | 59.3 | 82.1 | 66.4 | 51.7 | 64.4 |
| Mask2Former-VIS [3] | Swin-L | Full | 60.4 | **84.4** | 67.0 | – | – |
| MinVIS | Swin-L | Full | **61.6** | 83.3 | **68.6** | **54.8** | **66.6** |
| MinVIS | Swin-L | 1% | 59.0 | 81.6 | 64.7 | 54.0 | 64.0 |
| MinVIS | Swin-L | 5% | 59.3 | 81.4 | 65.8 | 53.8 | 64.1 |
| MinVIS | Swin-L | 10% | 61.0 | 83.0 | 67.7 | 54.6 | 66.1 |

Table 2: YouTube-VIS 2021 Results. MinVIS's performance improvement increases on the more challenging YouTube-VIS 2021. Our 1% results already outperform previous state-of-the-art.

| Method | Backbone | Training | AP | $AP_{50}$ | $AP_{75}$ | $AR_1$ | $AR_{10}$ |
|---|---|---|---|---|---|---|---|
| TeViT [8] | MsgShifT | Full | 37.9 | 61.2 | 42.1 | 35.1 | 44.6 |
| SeqFormer [5] | R50 | Full+C80k | 40.5 | 62.4 | 43.7 | 36.1 | 48.1 |
| Mask2Former-VIS [3] | R50 | Full | 40.6 | 60.9 | 41.8 | – | – |
| MinVIS | R50 | Full | **44.2** | **66.0** | **48.1** | **39.2** | **51.7** |
| SeqFormer [5] | Swin-L | Full+C80k | 51.8 | 74.6 | 58.2 | 42.8 | 58.1 |
| Mask2Former-VIS [3] | Swin-L | Full | 52.6 | 76.4 | 57.2 | – | – |
| MinVIS | Swin-L | Full | **55.3** | **76.6** | **62.0** | **45.9** | **60.8** |
| MinVIS | Swin-L | 1% | 52.9 | 74.9 | 58.9 | 44.7 | 58.3 |
| MinVIS | Swin-L | 5% | 54.3 | 76.3 | 60.1 | 45.4 | 59.5 |
| MinVIS | Swin-L | 10% | 54.9 | 76.3 | 61.9 | 45.3 | 60.1 |

previous state-of-the art. We also see that on YouTube-VIS datasets, reducing the annotations by 10x does not significantly affect the performances (-0.6% AP for 2019 and -0.4% AP for 2021).

**Occluded VIS (OVIS).** The results for OVIS are shown in Table 3. Mask2Former-VIS* denotes our application of Mask2Former-VIS to OVIS. Since videos in OVIS can have up to hundreds of frames, we apply Mask2Former-VIS to non-overlapping sliding windows of length 30. The outputs from these clips are then merged by our post-processing. MinVIS is an online method and does not need modification to apply to OVIS. MinVIS shows significant improvement compared to existing works on OVIS. With ResNet-50 backbone, MinVIS outperforms previous state-of-the-art TeViT with MsgShifT backbone by 7.6% AP. With Swin-L backbone, MinVIS outperforms previous best result MaskTrack R-CNN*+SWA by 10.5% AP, which is the winner of the 1st OVIS Challenge. Their key observation is that sampling frames that are far apart in OVIS leads to drastically different features and makes it hard to train MaskTrack R-CNN. This is in contrast to YouTube-VIS datasets, in which the videos are shorter and there are less temporal variations within the video. We observe the same phenomenon when training Mask2Former-VIS*. However, the limited sampling reference frame strategy of MaskTrack R-CNN*+SWA still does not work in this case. Mask2Former-VIS* uses a fully end-to-end loss instead of an explicit tracking loss to learn temporal association, which makes the learning even harder in OVIS. On the other hand, MinVIS is image-based and does not need to worry about the temporal sampling strategy to train the model. This is contrary to common belief that per-frame approaches are worse for scenarios with heavy occlusions. Instead, our image-based approach leads to easier and better learning on OVIS. We show that an image instance segmentation model that can segment occluded instances in each frame is also good at associating such instances

Table 3: OVIS Results. MinVIS significantly outperform existing approaches on OVIS. Our image-based framework leads to easier and better learning on this dataset with heavy occlusions.

| Method | Backbone | Training | AP | $AP_{50}$ | $AP_{75}$ | $AR_1$ | $AR_{10}$ |
|---|---|---|---|---|---|---|---|
| TeViT [8] | MsgShifT | Full | 17.4 | 34.9 | 15.0 | 11.2 | 21.8 |
| CrossVIS [2] | R50 | Full | 14.9 | 32.7 | 12.1 | 10.3 | 19.8 |
| CMaskTrack R-CNN [12] | R50 | Full | 15.4 | 33.9 | 13.1 | 9.3 | 20.0 |
| STC [33] | R50 | Full | 15.5 | 33.5 | 13.4 | 11.0 | 20.8 |
| Mask2Former-VIS* | R50 | Full | 17.3 | 37.3 | 15.1 | 10.5 | 23.5 |
| MinVIS | R50 | Full | **25.0** | **45.5** | **24.0** | **13.9** | **29.7** |
| MaskTrack R-CNN*+SWA [14] | Swin-L | Full | 28.9 | 56.3 | 26.8 | 13.5 | 34.0 |
| Mask2Former-VIS* | Swin-L | Full | 25.8 | 46.5 | 24.4 | 13.7 | 32.2 |
| MinVIS | Swin-L | Full | **39.4** | **61.5** | **41.3** | **18.1** | **43.3** |
| MinVIS | Swin-L | 1% | 31.7 | 54.9 | 31.3 | 16.3 | 36.1 |
| MinVIS | Swin-L | 5% | 35.7 | 60.1 | 35.8 | 17.3 | 39.9 |
| MinVIS | Swin-L | 10% | 37.2 | 60.7 | 38.0 | 17.3 | 41.1 |

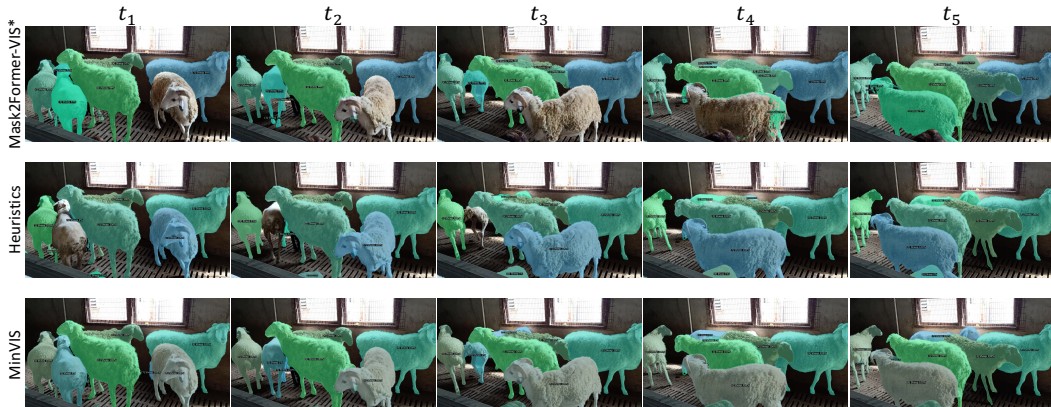

Figure 4: Qualitative results on OVIS. MinVIS stably tracks all the sheep in the video. Using mask overlap based heuristics instead leads to multiple identity switches in tracking. Mask2Former-VIS* uses per-clip training that is difficult to optimize on the challenging OVIS dataset.

across frames. Figure 4 shows qualitative results. MinVIS stably tracks all the sheep in the video. Using mask overlap based heuristics instead of query matching leads to multiple identity switches in tracking. Mask2Former-VIS* does not have as good segmentation masks because its training is interfered by heavy occlusions and large appearance deformations between frames in OVIS.

Figure 5 shows additional qualitative results on failure cases of MinVIS on the OVIS dataset. As discussed in Section 3.2, MinVIS does not use heuristics to handle the birth and death of object instances. The death of an object instance is correctly handled if its query is matched to a query in the next frame that produces an empty mask. Despite its simplicity and effectiveness, the drawback of this approach is that there is nothing stopping the model from matching the disappearing query to a query with a non-empty mask. From $t_3$ to $t_4$ in the top row of Figure 5, as the fish in the lower left leaves the frame, MinVIS associates it to a mask covering the tail of a nearby fish. From $t_4$ to $t_5$, when the fish in the upper left leaves the frame, MinVIS again associates it to a mask covering the head of a nearby fish. Since the associated masks are non-empty, MinVIS fails to correctly handle the death of these instances. On the other hand, when the two dogs in the bottom row of Figure 5 are covered in $t_2$, MinVIS correctly associates their queries to empty masks. MinVIS further correctly handles the object births in $t_3$. However, MinVIS is limited by the segmentation of the image segmentation model, which fails to segment the close-up person.

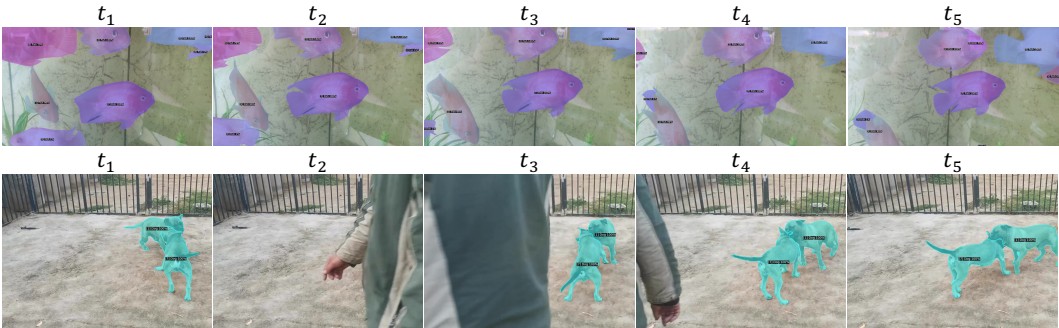

Figure 5: Failure cases of MinVIS on OVIS. When an object instance disappear from a video, MinVIS can fail by associating its query embedding to a wrong mask without overlap (top). This is because we do not use mask overlap heuristics in our work. On the other hand, we are also limited by the image instance segmentation model, which might not work well on close-up objects (bottom).

Table 4: Comparison of post-processing. Heuristics based on mask over laps lead to significant AP drop on OVIS. Our query matching approach has simpler design without loss of performance.

| Method | Dataset | AP | $AP_{50}$ | $AP_{75}$ | $AR_1$ | $AR_{10}$ |
|---|---|---|---|---|---|---|
| heuristics only | YouTube-VIS 2019 | 58.2 | 79.2 | 64.1 | 51.3 | 63.6 |
| heuristics + query | YouTube-VIS 2019 | 61.3 | 82.8 | **68.7** | 54.3 | 66.3 |
| query only | YouTube-VIS 2019 | **61.6** | **83.3** | 68.6 | **54.8** | **66.6** |
| heuristics only | YouTube-VIS 2021 | 52.7 | 75.3 | 57.3 | 44.4 | 58.3 |
| heuristics + query | YouTube-VIS 2021 | 55.1 | 76.2 | 61.9 | **46.0** | 60.7 |
| query only | YouTube-VIS 2021 | **55.3** | **76.6** | **62.0** | 45.9 | **60.8** |
| heuristics only | Occluded VIS | 31.7 | 56.0 | 31.3 | 15.8 | 35.8 |
| heuristics + query | Occluded VIS | 39.1 | **62.5** | 40.8 | 17.7 | **43.4** |
| query only | Occluded VIS | **39.4** | 61.5 | **41.3** | **18.1** | 43.3 |

## 4.2 Analyzing Query Matching

The success of MinVIS depends on whether query matching is good for tracking instances. We conduct ablation studies by comparing it to manually designed heuristics. We use the bipartite matching heuristics in Section 3.3 for tracking by treating instances in the last frame as groundtruth. The results are in Table 4. Using heuristics lead to around 3% AP drop on both YouTube-VIS 2019 and 2021. It leads to more significant drop on OVIS (7.7%) due to heavier occlusions. We also combine query matching and heuristics with equal weights, which has mixed results. Our query only approach is simpler and more generalizable without loss of performance.

We visualize the learned query embeddings by t-SNE [36] in Figure 3. Each plot is for a video in the training set. We visualize the training set to see the effect of an image only objective (to segment instances in an image) on query embeddings across different frames. Query embeddings of the same instance have the same color. We obtain the instance IDs for queries by bipartite matching its outputs to groundtruth instances, which have consistent IDs across frames. Without any video-based tracking objective, query embeddings of the same instances are already grouped into distinct clusters, even for the OVIS dataset. This supports our design of only using image-based objectives. In Appendix **??**, we further visualize query embeddings on videos not used in training.

## 4.3 Effect of Video-based Training

While we have shown that MinVIS achieves state-of-the-art VIS performance without video-based training, it is interesting to see how we can leverage video annotation when it is available. We use the video annotation to supervise our matching as in previous per-frame works [1, 2, 11]. Given two sampled frames, we use a hinge loss to ensure that the correct associations of queries have the highest inner products compared to that of other queries between the two frames [11]. The results are

Table 5: Results for adding supervision to query matching. The supervision can provide dataset dependent benefit if the temporal hyper-parameters are selected properly.

| Method | Dataset | AP | AP$_{50}$ | AP$_{75}$ | AR$_1$ | AR$_{10}$ |
|---|---|---|---|---|---|---|
| MinVIS | YouTube-VIS 2019 | **61.6** | **83.3** | **68.6** | **54.8** | **66.6** |
| + Supervised Matching | YouTube-VIS 2019 | 61.0 | 82.1 | 67.6 | 54.3 | 66.1 |
| + Limited Range | YouTube-VIS 2019 | 60.7 | 82.5 | 67.0 | 54.1 | 65.5 |
| MinVIS | YouTube-VIS 2021 | **55.3** | 76.6 | **62.0** | **45.9** | **60.8** |
| + Supervised Matching | YouTube-VIS 2021 | 54.4 | 75.7 | 60.6 | 45.5 | 59.5 |
| + Limited Range | YouTube-VIS 2021 | 55.2 | **77.0** | 61.5 | 45.4 | 60.1 |
| MinVIS | Occluded VIS | 39.4 | 61.5 | **41.3** | 18.1 | **43.3** |
| + Supervised Matching | Occluded VIS | 38.7 | 61.2 | 39.6 | 17.9 | 42.4 |
| + Limited Range | Occluded VIS | **39.6** | **63.2** | 41.0 | **18.2** | 43.0 |

in Table 5. The "Supervised Matching" rows mean directly applying the matching supervision to the original frame sampling process. In our case, this means that the two sampled frames might be separated up to 20 frames. As pointed out in previous works, frames that are far separated increase the training difficulty and can hurt model performances especially with occlusions [14]. We thus also consider the "Limited Range" training to only sample consecutive frames for supervised matching, as we only need to match consecutive frames. From the results, directly applying "Supervised Matching" hurt performances on all three datasets. Adding "Limited Range" recovers most of the performances for YouTube-VIS 2021 and OVIS. On OVIS, it even marginally outperforms the original MinVIS. However, this improvement does rely on the dataset dependent sampling range. We believe it is possible and important to use video-based training to further improve MinVIS, although this would take away MinVIS's practical advantages of only needing sparse annotations and having a simple training pipeline. Appendix **??** discusses further limitations of not using video information in training.

## 5 Conclusion

We show that a purely image-based training procedure can lead to competitive performances for VIS. Our key finding is that instance tracking naturally emerges in query-based image instance segmentation models with proper architectural constraints. In addition to improving state-of-the-art approaches on YouTube-VIS 2019/2021, we show that this is particularly beneficial for OVIS. The image-based objective reduces the learning difficulty and leads to better performances. MinVIS only requires sparse frame annotations, which makes it much more applicable to real-world scenarios. We believe a promising direction to extend MinVIS is to explore ways to better leverage the video frames that are not annotated to further improve our performances with sub-sampled annotations.

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
