# MinVIS: A Minimal Video Instance Segmentation Framework without Video-based Training

**De-An Huang**
NVIDIA
deahuang@nvidia.com

**Zhiding Yu**
NVIDIA
zhidingy@nvidia.com

**Anima Anandkumar**
Caltech, NVIDIA
anima@caltech.edu

## A  Limitations and Potential Negative Social Impacts

**Limitation.** We have discussed in the main paper on the possibility of improving MinVIS with video-based training. While we believe there are practical advantages of using our image-based VIS training pipeline, videos provide lots of extra information that we are not currently leveraging. In particular, temporal supervision from video should make our query embeddings even more suitable for tracking instances. In Figure 5, we visualize failure cases of using our current query embeddings for tracking. We conduct further analysis of Supervised Matching in Appendix F and believe further investigation along this direction should improve our approach. In addition to improving fully-supervised performance, we believe a promising direction is to explore semi-supervised learning at the frame level. In this case, one can temporally propagate the sub-sampled annotations in training to further improve the performance with reduced supervision.

**Potential Negative Social Impacts.** Video instance segmentation is a challenging video task, and thus provides fine-grained understanding of videos. The tracking and segmentation of objects of interest might be use for surveillance applications with negative social impact. While "person" is a category in the datasets used in this paper, no further protected attributes are annotated. Therefore, our trained models' performance on human subjects might not be fair with respect to protected attributes.

## B  Further Details for Datasets

The YouTube-VIS 2019/2021 datasets are under CC BY 4.0 License, and Occluded VIS is under CC BY-NC-SA 4.0 License. The videos in YouTube-VIS are from YouTube-VOS [37], whose videos are in turn from YouTube-8M [38]. YouTube-8M uses public videos on YouTube but does not discuss the process to filter personally identifiable information or offensive content in the paper.

## C  Tables with Standard Deviation

Tables with standard deviations are shown in Table 6, Table 7, and Table 8.

## D  Reducing Supervision for Mask2Former-VIS

The results for sub-sampling annotated frames for Mask2Former-VIS [3] are shown in Table 9. MinVIS consistently outperforms Mask2Former-VIS in all settings. The improvement increases for all three datasets when we sub-sample the annotation: +1.2% for full supervision v.s. +1.7% for 1% supervision on YouTube-VIS 2019. +2.7% for full supervision v.s. +5.8% for 1% supervision on YouTube-VIS 2021. +13.6% for full supervision v.s. +17.2% for 1% supervision on OVIS.

36th Conference on Neural Information Processing Systems (NeurIPS 2022).

Table 6: YouTube-VIS 2019 results. C80k indicates joint training with COCO images that have YouTube-VIS categories. MinVIS with X% means sub-sampling the annotated frames in training.

| Method | Backbone | Training | AP | $AP_{50}$ | $AP_{75}$ | $AR_1$ | $AR_{10}$ |
|---|---|---|---|---|---|---|---|
| TeViT [8] | R50 | Full | 42.1 | 67.8 | 44.8 | 41.3 | 49.4 |
| TeViT [8] | MsgShifT | Full | 46.6 | **71.3** | 51.6 | 44.9 | 54.3 |
| SeqFormer [5] | R50 | Full | 45.1 | 66.9 | 50.5 | 45.6 | 54.6 |
| SeqFormer [5] | R50 | +C80k | **47.4** | 69.8 | 51.8 | 45.5 | 54.8 |
| Mask2Former [3] | R50 | Full | 46.4 | 68.0 | 50.0 | – | – |
| MinVIS | R50 | Full | **47.4** ±0.2 | 69.0 ±2.1 | **52.1** ±0.2 | **45.7**±0.2 | **55.7** ±0.7 |
| TeViT [8] | Swin-L | Full | 56.8 | 80.6 | 63.1 | 52.0 | 63.3 |
| SeqFormer [5] | Swin-L | +C80k | 59.3 | 82.1 | 66.4 | 51.7 | 64.4 |
| Mask2Former [3] | Swin-L | Full | 60.4 | **84.4** | 67.0 | – | – |
| MinVIS | Swin-L | Full | **61.6**±0.3 | 83.3±0.2 | **68.6**±1.6 | **54.8**±0.4 | **66.6** ±0.9 |
| MinVIS | Swin-L | 1% | 59.0±0.3 | 81.6 ±0.4 | 64.7±1.3 | 54.0 ±0.3 | 64.0 ±0.4 |
| MinVIS | Swin-L | 5% | 59.3±0.2 | 81.4 ±1.7 | 65.8 ±0.7 | 53.8±0.4 | 64.1 ±0.2 |
| MinVIS | Swin-L | 10% | 61.0 ±0.7 | 83.0±0.8 | 67.7 ±1.8 | 54.6±0.3 | 66.1±0.1 |

Table 7: YouTube-VIS 2021 Results. MinVIS's performance improvement increases on the more challenging YouTube-VIS 2021. Our 1% results already outperform previous state-of-the-art.

| Method | Backbone | Training | AP | $AP_{50}$ | $AP_{75}$ | $AR_1$ | $AR_{10}$ |
|---|---|---|---|---|---|---|---|
| TeViT [8] | MsgShifT | Full | 37.9 | 61.2 | 42.1 | 35.1 | 44.6 |
| SeqFormer [5] | R50 | +C80k | 40.5 | 62.4 | 43.7 | 36.1 | 48.1 |
| Mask2Former [3] | R50 | Full | 40.6 | 60.9 | 41.8 | – | – |
| MinVIS | R50 | Full | **44.2**±0.3 | **66.0**±0.1 | **48.1**±0.7 | **39.2** ±0.3 | **51.7** ±0.7 |
| SeqFormer [5] | Swin-L | +C80k | 51.8 | 74.6 | 58.2 | 42.8 | 58.1 |
| Mask2Former [3] | Swin-L | Full | 52.6 | 76.4 | 57.2 | – | – |
| MinVIS | Swin-L | Full | **55.3**±0.2 | **76.6**±0.3 | **62.0**±0.8 | **45.9**±0.2 | **60.8** ±0.3 |
| MinVIS | Swin-L | 1% | 52.9 ±0.4 | 74.9±0.5 | 58.9 ±0.7 | 44.7 ±0.3 | 58.3 ±0.7 |
| MinVIS | Swin-L | 5% | 54.3 ±0.3 | 76.3 ±0.5 | 60.1±0.3 | 45.4 ±0.4 | 59.5 ±0.2 |
| MinVIS | Swin-L | 10% | 54.9±0.3 | 76.3±0.6 | 61.9±0.2 | 45.3 ±0.2 | 60.1 ±0.4 |

Table 8: OVIS Results. MinVIS significantly outperform existing approaches on OVIS. Our image-based framework leads to easier and better learning on this dataset with heavy occlusions.

| Method | Backbone | Training | AP | $AP_{50}$ | $AP_{75}$ | $AR_1$ | $AR_{10}$ |
|---|---|---|---|---|---|---|---|
| TeViT [8] | MsgShifT | Full | 17.4 | 34.9 | 15.0 | 11.2 | 21.8 |
| CrossVIS [2] | R50 | Full | 14.9 | 32.7 | 12.1 | 10.3 | 19.8 |
| CMaskTrack R-CNN [12] | R50 | Full | 15.4 | 33.9 | 13.1 | 9.3 | 20.0 |
| STC [33] | R50 | Full | 15.5 | 33.5 | 13.4 | 11.0 | 20.8 |
| Mask2Former-VIS* | R50 | Full | 17.3 | 37.3 | 15.1 | 10.5 | 23.5 |
| MinVIS | R50 | Full | **25.0**±0.3 | **45.5**±0.6 | **24.0**±0.7 | **13.9**±0.3 | **29.7**±0.3 |
| MaskTrack R-CNN*+SWA [14] | Swin-L | Full | 28.9 | 56.3 | 26.8 | 13.5 | 34.0 |
| Mask2Former-VIS* | Swin-L | Full | 25.8 | 46.5 | 24.4 | 13.7 | 32.2 |
| MinVIS | Swin-L | Full | **39.4**±0.5 | **61.5**±0.1 | **41.3**±0.6 | **18.1**±0.1 | **43.3**±0.5 |
| MinVIS | Swin-L | 1% | 31.7±0.5 | 54.9 ±1.0 | 31.3±0.5 | 16.3±0.3 | 36.1±0.3 |
| MinVIS | Swin-L | 5% | 35.7±0.4 | 60.1 ±1.2 | 35.8±0.7 | 17.3±0.1 | 39.9±0.3 |
| MinVIS | Swin-L | 10% | 37.2±0.5 | 60.7 ±1.1 | 38.0±1.0 | 17.3±0.2 | 41.1±0.4 |

# E  Visualizing Query Embeddings in Evaluation

In the main paper, we visualize the learned query embeddings by t-SNE [36] in Figure 3. The videos in Figure 3 are in the training set and the figure is meant to understand how query embeddings in training cluster by instances without video-based loss function. We can similarly apply the same visualization to videos that are not used in training. One complication here is that this visualization uses groundtruth instance annotations to determine the corresponding instance ID for each query. However, the groundtruth annotation is not publicly available for the three datasets considered in this work. Our reported results are obtained by submitting our predictions to the datasets' evaluation servers. We therefore perform this analysis by training a new model that only uses 90% of the training videos in YouTube-VIS 2019, and visualize the learned model's query embeddings during evaluation on the 10% videos that are not used to train the model. While the 10% videos are not used in training the model, we still have their groundtruth instances for visualization purposes. This

Table 9: Sub-sampling the annotated training frames for MinVIS and Mask2Former-VIS. MinVIS outperforms Mask2Former-VIS for all of our settings. The improvement of MinVIS increases as we further sub-sample the annotated frames.

| Method | Dataset | Backbone | Full | 10% | 5% | 1% |
|---|---|---|---|---|---|---|
| Mask2Former-VIS | YTVIS-19 | Swin-L | 60.4 | 59.0 | 57.8 | 57.3 |
| MinVIS | YTVIS-19 | Swin-L | **61.6** | **61.0** | **59.3** | **59.0** |
| Mask2Former-VIS | YTVIS-21 | Swin-L | 52.6 | 51.2 | 50.0 | 47.1 |
| MinVIS | YTVIS-21 | Swin-L | **55.3** | **54.9** | **54.3** | **52.9** |
| Mask2Former-VIS | OVIS | Swin-L | 25.8 | 24.1 | 22.3 | 14.5 |
| MinVIS | OVIS | Swin-L | **39.4** | **37.2** | **35.7** | **31.7** |

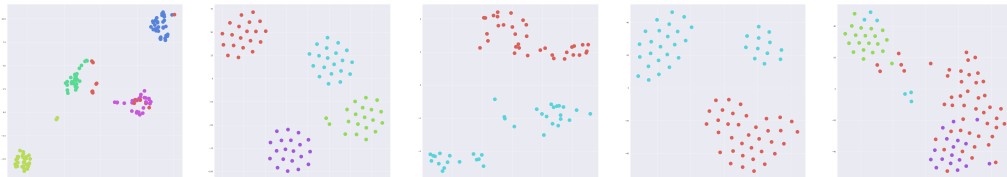

Figure 6: Visualizing our query embeddings during evaluation on videos not used in training. Each plot is for a video, and query embeddings of the same instance (from different frames) have the same color. Despite being noisier than training videos, the query embeddings are still grouped into clusters by instance without any video-based training.

provides a realistic approximation of how our query embeddings would look like for videos not used in training. The visualization is in Figure 6. Despite being noisier than training videos, the query embeddings are still grouped into clusters by object instances without any video-based training. This is also quantitatively supported by our state-of-the-art VIS performance on the three datasets.

## F   Further Analysis of Supervised Matching

We conduct further analysis on the results in Section 4.3. We visualize the query embeddings on the same training videos with and without using supervised matching. In particular, we perform the

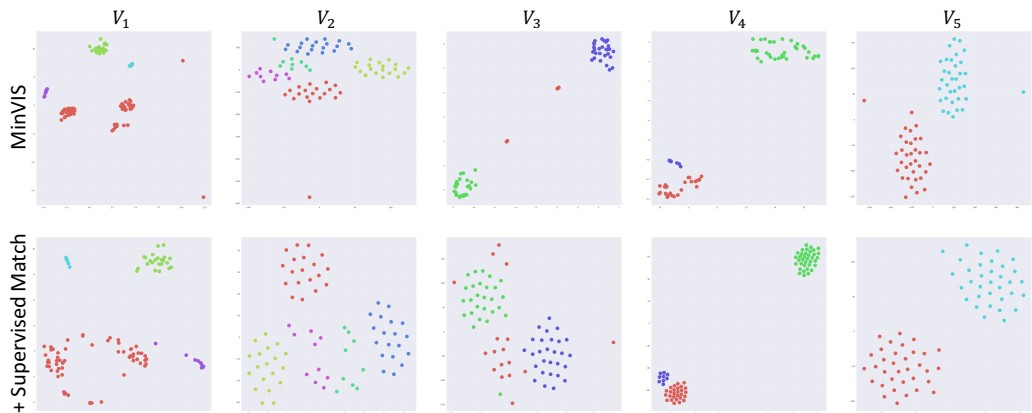

Figure 7: Visualizing learned query embeddings on the same videos with and without Supervised Matching. Plots in the same column visualize the same video $V_i$. Supervised matching makes the embeddings more evenly distributed and smooths out the outliers in the embedding space. However, it is unclear whether this is overall beneficial to our tracking by query matching.

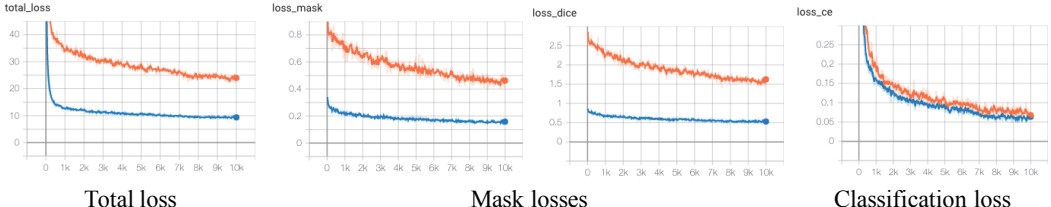

| Total loss | Mask losses | Classification loss |

Figure 8: Comparing the training curves of MinVIS and Mask2Former-VIS on OVIS. Blue curves are MinVIS and orange curves are Mask2Former-VIS.

analysis on YouTube-VIS 2019 and compare MinVIS v.s. MinVIS + Supervised Matching + Limited Range, which hurts performance the most in Table 5. The visualizations are in Figure 7. While the plots look similar for most videos, one consistent trend we observe is that adding supervised matching makes the embeddings more evenly distributed and smooths out the outliers in the embedding space. This is a reasonable consequence as the objective encourages the embeddings from the same object instance to be closer to each other. However, it is unclear whether this is overall beneficial to our tracking by query matching. For example, in $V_3$, the outliers are removed at the cost of mixing embeddings from different instances. We believe it is an important future work to further understand how to better leverage video information to improve MinVIS.

## G    Baseline Training Curves on OVIS

As discussed in the main paper, it is difficult to optimize our per-clip baseline on the challenging OVIS dataset. We have included the training curves in Figure 8 for further illustration. Blue curves are MinVIS and orange curves are Mask2Former-VIS. While the classification loss still optimizes well on OVIS, the per-clip baseline has difficulty optimizing mask related losses.