# OpenReview forum: "MinVIS: A Minimal Video Instance Segmentation Framework without Video-based Training"
_NeurIPS.cc/2022/Conference — NeurIPS 2022 Accept_

### Official Review · Reviewer_zDMq · 2022-07-06

**Rating:** 7
**Confidence:** 4
**Soundness:** 4 excellent
**Presentation:** 3 good
**Contribution:** 3 good

**Summary:**

The paper proposes MinVIS, a minimal video instance segmentation system (VIS) that obtains SOTA results on a variety of VIS benchmarks without employing video-based architectures or video-based training. The proposed system operates in two stages: 1) first, an image-level instance segmentation is trained, 2) afterward, the frame-level instance associations across frames are built using a bipartite matching algorithm. The authors demonstrate that despite not using any video-related information, their proposed framework achieves highly competitive results in comparison to video-based VIS solutions.

**Questions:**

Questions:
1. Could the authors briefly summarize their TECHNICAL contributions, and also present a detailed comparison with the Mask2Former system, and how their proposed approach differs from this prior work.
2. I was quite puzzled by the results obtained using video-based training in 4.3. It's quite surprising to see the performance drop, and the justification that this happens due to occlusions doesn't seem very convincing to me. This is based on my own past experiences, where such pairwise matching-based training was highly successful for tracking. It would be useful if the authors could visualize the embeddings resulting from such training as they did in Figure 3 of the original draft.
3. It would also be useful if the authors could incorporate some efficiency metrics. I imagine that the proposed approach is more efficient than prior video-based systems. It would be good to highlight such advantages.

**Limitations:**

In my view, the authors could discuss the limitations of not incorporating any video-based architectures/training procedures in more depth. Video provides lots of extra information that the authors are ignoring in this case. While I appreciate the simplicity and effectiveness of their solution, I think it would also be useful to have a more detailed discussion how their framework could be improved while incorporating video cues. Section 4.3 does not provide a good discussion on that.

**Strengths And Weaknesses:**

Strengths:
+ The authors tackle a challenging and highly impactful problem of video instance segmentation (VIS).
+ The proposed system is very simple, which would provide a convenient baseline for other researchers to build on.
+ The experiments are thorough and convincing, i.e., the proposed method achieves state-of-the-art results on multiple VIS benchmarks.
+ The paper is easy to read and understand.
+ Additional experiments showing strong performance with a limited amount of labeled VIS data.
+ The proposed method can be used in an online setting.

Weaknesses:
- Overall, I enjoyed reading the paper. The proposed approach is simple, yet highly effective. My only concern is that the proposed approach feels somewhat incremental, i.e., the technical contributions are quite limited. To be more precise, the proposed framework is largely built on Mask2Former, which was also originally developed for images. The bipartite matching algorithm to associate instances across frames has been widely used both in the image-only settings and in the video-related settings in the past. The only substantial contribution that I could spot is in the design of a prediction head, i.e., imposing the constraint to have Q convolve with the whole feature map F_{−1}. It's a small, yet important design choice that simplifies instance segmentation prediction.
- In my view, the authors could discuss the limitations of not incorporating any video-based architectures/training procedures in more depth. Video provides lots of extra information that the authors are ignoring in this case. While I appreciate the simplicity and effectiveness of their solution, I think it would also be useful to have a more detailed discussion how their framework could be improved while incorporating video cues. Section 4.3 does not provide a good discussion on that.
- The authors should carefully proofread their draft. There are some typos.

---

> ### Author Response · Authors · 2022-08-02
> **Author Response**
>
> Thank you for your comments. We have revised the paper based on your comments. Please find our responses to specific questions below.
>
>  - **Technical Contribution**: We agree with the reviewer that it is important to clarify our technical contribution. We would like to clarify that our approach is not limited to Mask2Former as pointed out by the other reviewer. Our key finding is that instance tracking naturally emerges in image instance segmentation models with proper architectural constraints. It is non-trivial to find that video-based architecture and training are not required for competitive VIS performance. This finding is especially important when most of the recent approaches in VIS are dominated by the per-clip paradigm. We further leverage this finding to achieve state-of-the-art VIS performance on multiple datasets (over 10% improvement on OVIS) by only training an image instance segmentation model. Our approach thus directly brings advances in image instance segmentation to the video space and has practical advantages including reducing required supervision in our experiments (can use only 1% of labeled frames).
>
>  - **Visualizing Embeddings for Supervised Matching**: We agree with the reviewer that it is important to further analyze supervised matching in Section 4.3. We conduct the same visualization for MinVIS + Supervised Matching + Limited Range on YouTube-VIS 2019. More details are in Appendix G. While the plots look similar for most videos, one consistent trend we observe is that adding supervised matching makes the embeddings more evenly distributed and smooths out the outliers in the embedding space. Some examples are shown in Figure 7. This is a reasonable consequence as the objective encourages the embeddings from the same object instance to be closer to each other. However, it is unclear whether this is overall beneficial to our tracking by query matching. For example, in $V_3$ in Figure 7, the outliers are removed at the cost of mixing embeddings from different instances.
>
>  - **Efficiency Benefits of MinVIS**: We agree with the reviewer that our image-based approach should have computational advantages. The key difference compared to per-clip approaches is that now our Transformer decoder only needs to process tokens in an image, instead of having a complexity that’s quadratic with respect to the number of frames. We conduct further benchmarking and find that the Transformer decoder component only accounts for less than 10% of the computational time in our current implementation. A large part of the computation is in the visual backbone processing of each frame, which is the same for our approach and per-clip baselines. Thus, we do not observe significant speed improvements. However, our approach does have a significant memory advantage especially for long videos, as our memory complexity is in principle independent of video length.
>
>  - **Limitations of Not Using Video-based Training**: We agree with the reviewer that videos provide lots of extra information that we are not currently leveraging. We have expanded Appendix A to discuss limitations of not incorporating any video-based architectures/training procedures in more depth. More specifically, temporal supervision should improve our tracking performance to prevent failure cases in Appendix F. In addition, video information is also beneficial to explore semi-supervised settings, where we temporally propagate the annotation to improve the performance of our model trained with sub-sampled annotations.

---

### Official Review · Reviewer_o2a1 · 2022-07-09

**Rating:** 5
**Confidence:** 5
**Soundness:** 4 excellent
**Presentation:** 3 good
**Contribution:** 3 good

**Summary:**

This paper proposes a new video instance segmentation (VIS) framework, MinVIS, which achieves state-of-the-art VIS performance with neither video-based architectures nor video-based training procedures. With this image-based framework, this work is more flexible on the requirements of training data.

**Questions:**

As described in weakness.

**Limitations:**

No serious negative societal impact in this work.

**Strengths And Weaknesses:**

Strengths:
1) This paper achieves SOTA results on various video instance segmentation datasets.
2) The method of this paper is simple but satisfied in performance.
3) This paper is well organized.

Weakness:
1) The technical contribution of this paper is a bit sufficient. The critical contribution of this paper is to utilize the query embedding from Mask2Former [27] and prove it is effective for instance association, but no more novel idea or method is proposed.

---

> ### Author Response · Authors · 2022-08-02
> **Author Response**
>
> Thank you for your comments. Please find our response to your question below.
>
>  - **Technical Contribution**: We agree with the reviewer that our approach uses Mask2Former’s query embedding for instance association and it is important to clarify our contribution. We would like to clarify that our approach is not limited to Mask2Former as pointed out by the other reviewer. Our key finding is that instance tracking naturally emerges in image instance segmentation models with proper architectural constraints. It is non-trivial to find that video-based architecture and training are not required for competitive VIS performance. This finding is especially important when most of the recent approaches in VIS are dominated by the per-clip paradigm. We further leverage this finding to achieve state-of-the-art VIS performance on multiple datasets (over 10% improvement on OVIS) by only training an image instance segmentation model. Our approach thus directly brings advances in image instance segmentation to the video space and has practical advantages including reducing required supervision in our experiments (can use only 1% of labeled frames).

---

### Official Review · Reviewer_2Rax · 2022-07-11

**Rating:** 8
**Confidence:** 5
**Soundness:** 4 excellent
**Presentation:** 4 excellent
**Contribution:** 3 good

**Summary:**

The paper proposes a novel video instance segmentation framework called Min-Vis which leverages transformed based architecture to segment object instances in videos. The key contributions are:

* The paper demonstrates that good performance can be obtained for this problem even by using image level instance segmentation annotations alone. And instance tracking problem can be solved by conducting bipartite matching on object queries between adjacent frames without requiring any training.

* The paper also demonstrates that because of the simplicity of its design, it can even be trained with much smaller dataset size.

* The paper achieves state-of-the-art performance on multiple VIS benchmarks.

**Questions:**

More details on compute times for training and inference will be interesting to learn about.

**Limitations:**

Yes

**Strengths And Weaknesses:**

Strengths:

* This is a solid paper with state-of-the-art results for the problem of video instance segmentation. The gains on YouTube-VIS 2021 and OVIS dataset are impressive. The paper also presents good ablation studies. Looking at the current leaderboard for YouTube-VIS dataset, the proposed method looks to be the best.

* The proposed approach is fairly generic and can be extended to other tasks beyond instance segmentation. More experiments are needed but if the assumption that cosine similarity between object query embeddings is enough to associate instances between frames holds, this will be a natural method to apply on multi-object tracking problems as well.

* The ability of being able to train video instance segmentation models using image only instance segmentation masks brings a significant practical advantage to this method compared to others.

Weakness:
* More datasets like DAVIS benchmark and "Unidentified Video Objects" from Meta can be used for additional experiments. This will provide more diversity to the benchmarks on which the method is evaluated.
* No analysis on model failures are presented in the paper, a brief discussion with some visual examples could provide more insights into where the current approach breaks down.
* Visualization of object embeddings (Figure 3) should be presented for validation sets as well for better understanding.

---

> ### Author Response · Authors · 2022-08-02
> **Author Response**
>
> Thank you for your comments. We have revised the paper based on your comments. Please find our responses to specific questions below.
>
>  - **Qualitative Results on Failure Cases**: We agree with the reviewer that it is important to provide analysis on model failures in our paper. The additional results and discussions are in Figure 6 and Appendix F. As discussed in Section 3.2, MinVIS does not use any heuristics to handle the birth and death of object instances. The death of an object instance is correctly handled if its query is matched to a query in the next frame that produces an empty mask. Despite its simplicity and effectiveness, the drawback of this approach is that there is nothing stopping the model from matching the disappearing query to a query with a non-empty mask. The top of Figure 6 shows an example when two fishes on the left leave the frame, MinVIS associates them with non-empty masks on nearby fishes. On the other hand, while MinVIS correctly handles the object birth and death in the bottom row, MinVIS is still limited by the image segmentation model, which fails to segment the close-up person.
>
>  - **t-SNE Visualization on Validation Sets**: We agree with the reviewer that visualizing embeddings in validation provides better understanding of our approach. The visualization is in Figure 5. Since our visualization uses groundtruth, which is not publicly available for validation sets of the datasets used in this paper, we further use a 90/10 split on the training set for this visualization. The details are in Appendix E. Despite being noisier than training videos, the query embeddings are still grouped into clusters by object instances for videos not used in training. This is also quantitatively supported by our state-of-the-art VIS performance on the three datasets.
>
>  - **Compute Times**: Since our models are initialized from COCO-pretrained weights, we only need to fine-tune for 6k iterations on YouTube-VIS 2019 (10k for OVIS). The whole training takes about 1 hour for R50 on 8 V100 GPUs, and about 1.25 hour for Swin-L on 16 V100 GPUs. For inference, each video on average takes 0.9s for R50 and 2.0s for Swin-L

---

### Official Review · Reviewer_tb44 · 2022-07-12

**Rating:** 7
**Confidence:** 4
**Soundness:** 3 good
**Presentation:** 3 good
**Contribution:** 3 good

**Summary:**

The paper presents a video instance segmentation method for simultaneously segmenting and tracking objects in video sequences. The proposed per-frame method first uses the query-based MaskFormer [27] to segment objects frame by frame, and associate object segments in different frame by matching the queries. The proposed method achieves SOTA performance on three VIS datasets. Additionally, the paper demonstrates competitive results of the proposed method when 1%, 5% and 10% of the training data is available.

**Questions:**

As mentioned in weaknesses above, what are the performance of baselines with 1%, 5%, 10% of the training data? Preferably data augmentation should be performed. Note that the clip-based baseline MaskFormer-VIS uses clips consisting of only 2 frames for training. Therefore, one can generate pairs of frames for training using data augmentation even in the 1% setting

What are the data augmentation strategies for training MaskFormer in the proposed approach, especially in the low data settings? Is the data augmentation performed in the low data settings different from the one with full data?

Why don’t  Table 4 and 5 have the standard deviation included? It seems hard to conclude which strategies are better as the APs for some of them are very similar (within 0.2 difference).

According to Ln 258, the temporal sampling strategy during training is important. What is the sampling strategy for the baselines? Do the authors try different sampling strategies and data augmentation on OVIS dataset except for those default ones on YouTube-VIS?

In Fig 4 column $t_1$ to $t_4$, why does the baseline Mask2Former-VIS miss to detect the second sheep from the right in the image? It doesn't make a lot of sense to me that MinVIS can detect that object but Mask2Former fails given that both of them are highly based on MaskFormer. Is it because Mask2Former-VIS uses a single query for an object in the video, and using only a single query cannot adapt to the drastic appearance change in this case? If it is the case, the choice of applying Mask2Former-VIS to clips with length of 30 (Ln 246) might not be optimal.

**Limitations:**

Yes, the authors addressed the limitations and potential negative societal impact in section 4.3 and the supplementary document.

**Strengths And Weaknesses:**

**Strengths**

The proposed approach takes the advantage of the design of the query-based image segmentation method and extend it to tracking object segments effectively. The proposed method achieves SoTA performance across three VIS datasets.

The proposed method only requires training the query-based image segmentation method. I appreciate the simplicity.

The paper is easy to follow.

****

**Weaknesses**

The key idea of the proposed approach is to use queries to associate object segments in different frame. In terms of originality, this is not entirely novel and have been used and studied in other related tasks, e.g., multiple object tracking (MOT). Importantly, the paper ignores literature review on related query-based works that also use queries for object association and tracking or related video tasks.

In the experiments, the authors show competitive results of their approach with little amount of training data. However, it’s unclear whether other baselines also have little reduction in the performance even with little amount of data. Note that data augmentation can and should be applied for those baselines, and it is still a fair comparison in terms of the amount of training data.

The method highly depends on MaskFormer’s architecture design and its segmentation results.

---

> ### Author Response · Authors · 2022-08-02
> **Author Response**
>
> Thank you for your comments. We have revised the paper based on your comments. Please find our responses to specific questions below.
>
>  - **Data Augmentation**: We use standard data augmentation strategies as in previous works: randomly resizing shortest edge to [360, 480], and randomly flipping the video clip. We use the same data augmentation for all settings (1%, 5%, 10%, Full Supervision).
>
>  - **Reduced Supervision for Baseline**: We agree with the reviewer that it is important to also apply our annotation sub-sampling experiment to per-clip baselines. We thus also apply our low data settings to Mask2Former-VIS. The full results are in Table 9 and Appendix D. MinVIS consistently outperforms Mask2Former-VIS in all settings. The improvement increases for all three datasets when we  sub-sample the annotation: +1.2% for full supervision v.s. +1.7% for 1% supervision on YouTube-VIS 2019. +2.7% for full supervision v.s. +5.8% for 1% supervision on YouTube-VIS 2021. +13.6% for full supervision v.s. +17.2% for 1% supervision on OVIS.
>
>  - **Tables with Standard Deviation**: Tables with standard deviation are in Appendix C, which is originally in the supplementary. We have now moved the appendices from supplementary to after the main paper.
>
>  - **Temporal Sampling on OVIS for Baseline**: The reported results on OVIS use default temporal sampling on YouTube-VIS. However, we have also experimented with the limited range strategy for Mask2Former-VIS, which did not improve the results. Therefore, we report the default temporal sampling result for consistency. As discussed in “Implementation Details” the only hyperparameter change for OVIS is to increase the number of iterations from 6k to 10k.
>
>  - **Figure 4 Explanation**: We agree with the reviewer that Figure 4 requires further explanation. While both MinVIS and Mask2Former-VIS are built on the Mask2Former architecture, this does not imply that both would have the same behavior because even the same architecture can still learn different model weights during training. The difference in training is that Mask2Former-VIS’s training objective asks the model to directly resolve tracking under heavy occlusion. This is hard to optimize and affect the model performance. We have included the training curves comparing Mask2Former-VIS and MinVIS in Appendix H for further illustration. Mask2Former-VIS has a much higher total loss compared to MinVIS in training, which affects the segmentation performance in inference.

---

> > ### Comment · Reviewer_tb44 · 2022-08-09
> > **Post rebuttal rating - Accept**
> >
> > I would like to thank the authors for their responses. Most of my concerns are addressed. I’ve read the comments from other reviewers and the author responses. I’ve also read the revised manuscript and appendix (the paragraphs highlighted in blue). The additional experiments and qualitative results for failure cases are useful analysis.
> >
> > Overall, the paper proposes a neat idea that works effectively on common VIS benchmarks.  I think it’s a nice paper and will be of interest to the community. I therefore recommend to accept the paper.

---

> > > ### Author Response · Authors · 2022-08-09
> > > **Thank You**
> > >
> > > Dear reviewer, thank you very much for the appreciation of this work!

---

### Meta-Review · Area_Chair_79an · 2022-08-27

**Recommendation:** Accept
**Confidence:** Certain

**Metareview:**

All four reviewers are positive about this work. Reviewers appreciate the clear writing, simple yet highly effective idea, and strong experimental validation on three Video Instance Segmentation datasets. The authors responses further clarified and sufficiently addressed the concerns from the reviewers. The AC reads the reviews, the rebuttal, and agree with the reviewers to recommend acceptance.

**Award:**

No

---

### Decision · Program_Chairs · 2022-09-14

Accept